# The sound of a Martian dust devil

N. Murdoch [1] ✉, A. E. Stott [1], M. Gillier[1], R. Hueso [2], M. Lemmon[3], G. Martinez [4,5], V. Apéstigue [6], D. Toledo [6], R. D. Lorenz [7], B. Chide[8], A. Munguira [2], A. Sánchez-Lavega [2], A. Vicente-Retortillo [9], C. E. Newman [10], S. Maurice[11], M. de la Torre Juárez [12], T. Bertrand [13], D. Banfield [14,15], S. Navarro [9], M. Marin[9], J. Torres[9], J. Gomez-Elvira [6], X. Jacob [16], A. Cadu[1], A. Sournac[1], J. A. Rodriguez-Manfredi [9], R. C. Wiens [17] & D. Mimoun [1]

Dust devils (convective vortices loaded with dust) are common at the surface of Mars, particularly at Jezero crater, the landing site of the Perseverance rover. They are indicators of atmospheric turbulence and are an important lifting mechanism for the Martian dust cycle. Improving our understanding of dust lifting and atmospheric transport is key for accurate simulation of the dust cycle and for the prediction of dust storms, in addition to being important for future space exploration as grain impacts are implicated in the degradation of hardware on the surface of Mars. Here we describe the sound of a Martian dust devil as recorded by the SuperCam instrument on the Perseverance rover. The dust devil encounter was also simultaneously imaged by the Perseverance rover's Navigation Camera and observed by several sensors in the Mars Environmental Dynamics Analyzer instrument. Combining these unique multi-sensorial data with modelling, we show that the dust devil was around 25 m large, at least 118 m tall, and passed directly over the rover travelling at approximately 5 m s$^{-1}$. Acoustic signals of grain impacts recorded during the vortex encounter provide quantitative information about the number density of particles in the vortex. The sound of a Martian dust devil was inaccessible until SuperCam microphone recordings. This chance dust devil encounter demonstrates the potential of acoustic data for resolving the rapid wind structure of the Martian atmosphere and for directly quantifying wind-blown grain fluxes on Mars.

[1]Institut Supérieur de l'Aéronautique et de l'Espace (ISAE-SUPAERO), Université de Toulouse, Toulouse, France. [2]Física Aplicada, Escuela de Ingeniería de Bilbao, Universidad del País Vasco (UPV/EHU), Bilbao, Spain. [3]Space Science Institute, Boulder, CO 80301, USA. [4]Lunar and Planetary Institute, Universities Space Research Association, Houston, TX, USA. [5]Department of Climate and Space Sciences and Engineering, University of Michigan, Ann Arbor, MI, USA. [6]Instituto Nacional de Técnica Aeroespacial, Madrid, Spain. [7]Space Exploration Sector, Johns Hopkins Applied Physics Laboratory, Laurel, MD, USA. [8]Space and Planetary Exploration Team, Los Alamos National Laboratory, Los Alamos, NM, USA. [9]Centro de Astrobiología (INTA-CSIC), Madrid, Spain. [10]Aeolis Research, Chandler, AZ, USA. [11]Institut de Recherche en Astrophysique et Planétologie, Université de Toulouse 3 Paul Sabatier, CNRS, CNES, Toulouse, France. [12]Jet Propulsion Laboratory, California Institute of Technology, Pasadena, CA, USA. [13]Laboratoire d'Etudes Spatiales et d'Instrumentation en Astrophysique (LESIA), Observatoire de Paris, Université PSL, CNRS, Sorbonne Université, Univ. Paris Diderot, Sorbonne Paris Cité, 5 place Jules Janssen, 92195 Meudon, France. [14]Cornell University, Ithaca, NY, USA. [15]NASA AMES Research Center, Moffett Field, CA, USA. [16]Institut de Mécanique des Fluides, Université de Toulouse III Paul Sabatier, INP, CNRS, Toulouse, France. [17]Earth, Atmospheric, and Planetary Sciences, Purdue University, West Lafayette, IN, USA. ✉e-mail: naomi.murdoch@isae-supaero.fr

Convective vortices and dust devils (convective vortices loaded with dust) are common at the surface of Mars[1–6] and have been found to be abundant at Jezero crater[7,8], the NASA Mars 2020 Perseverance rover landing site on Mars (18.363°N, 77.595°E, elevation −2656 m). Forming typically at the corners and edges of convection cells, such vortices occur when warm air close to the surface starts to rise and begins to rotate, in turn generating a pressure depression in the center of the vortex and strong rotational winds[9]. Convective vortices are frequent on Mars' surface during the day, and constitute a key element of the Planetary Boundary Layer (PBL), the lowest part of the atmosphere in direct contact with the planetary surface. The PBL plays an important role in the mixing of heat, momentum, dust, and chemical species[10]. Therefore, understanding its dynamics[11–13] is a crucial step in understanding the meteorology of a planet.

Convective vortices are also indicators of atmospheric turbulence (larger numbers of convective vortices occur during the most turbulent periods of the day)[14] and are an important lifting mechanism for the Martian dust cycle[12,15,16]. However, not all convective vortices lift dust and it is not clear why Jezero crater has particularly abundant dust devils[7], whereas the cameras[17] on board the NASA InSight lander did not directly image a single dusty vortex in western Elysium Planitia (4.502°N, 135.623°E, −2613 m) during more than a full Martian year. Grain transport drives the erosion rates and geomorphology of the Martian surface, and dust abundance in the atmosphere determines much of the radiative and thermal properties of the atmosphere, which in turn strongly influence circulation. Understanding the particle flux associated with the grain lifting processes is, therefore, a matter of crucial importance for aeolian research. An improved understanding of surface lifting and atmospheric transport is also required for accurate simulation of the dust cycle and prediction of dust storms in Mars climate models[16]. The extent of observed dust lifting and sand motion on Mars is much greater than that expected from modeled and observed near-surface winds, based on the thresholds for grain motion predicted by theory, numerical grain motion modeling, and most laboratory experiments under Mars-like conditions[18,19]. Explanations for the apparently lower-than-expected thresholds have been

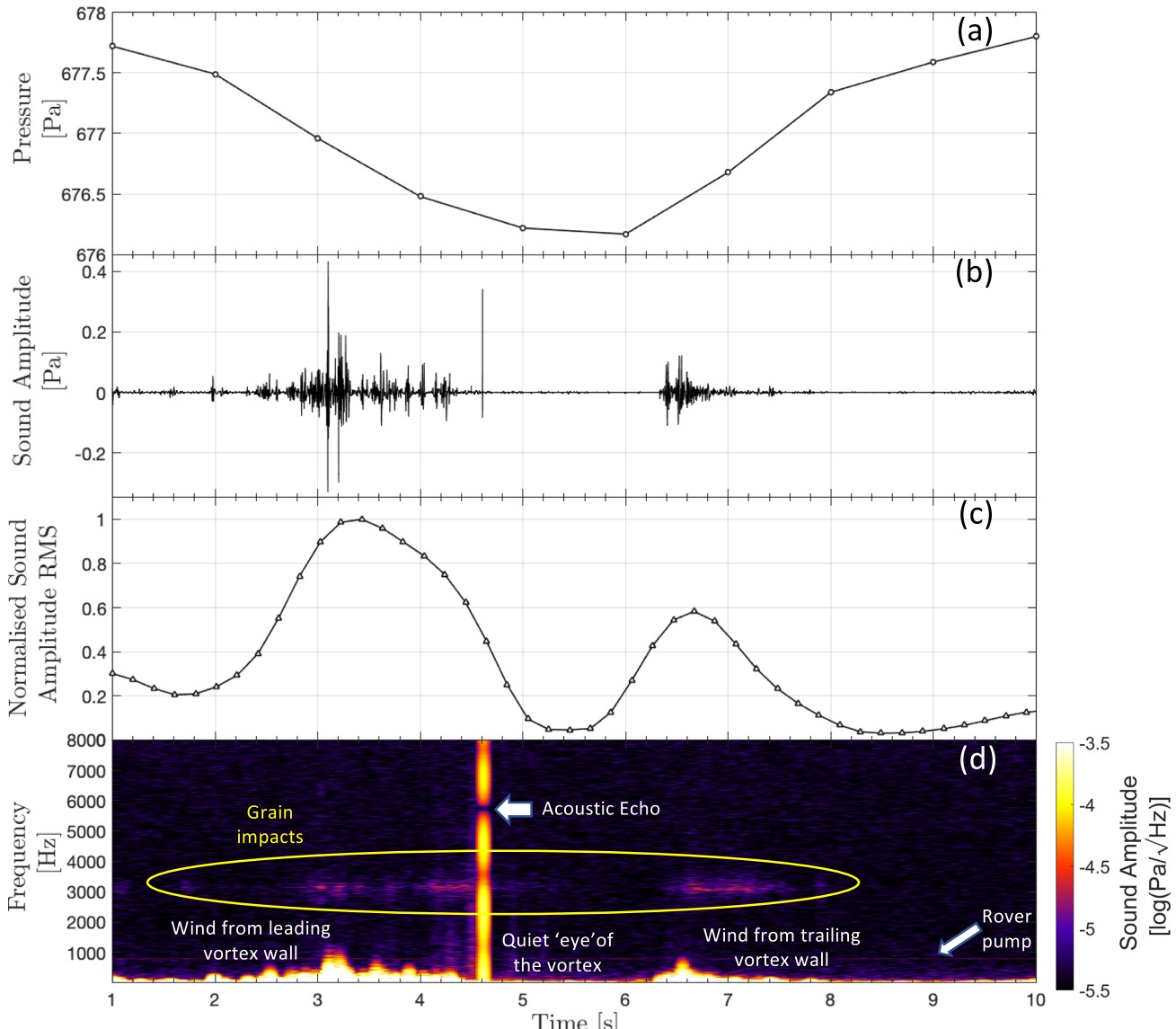

**Fig. 1 | Acoustic data during a dusty vortex encounter. a** The pressure data during the dust devil encounter at 11.02 LTST on September 27, 2021 (Perseverance sol 215), **b** the SuperCam microphone sound amplitude as a function of time, **c** the normalised Root-Mean-Square (RMS) of the microphone signal in the 20–60 Hz bandwidth (the region containing the atmospheric signal[37,38]) calculated in 2-s windows, **d** the spectrogram of the microphone sound pressure level (calculated using a window size of 0.2 s in order to resolve the fine details in the acoustic data) showing the wind noise, the grain impacts and also the rover pump harmonic at 760 Hz, and the acoustic echo at ~6 kHz due to sound reflections from the base of the microphone[37]. Source data are provided as a Source Data file.

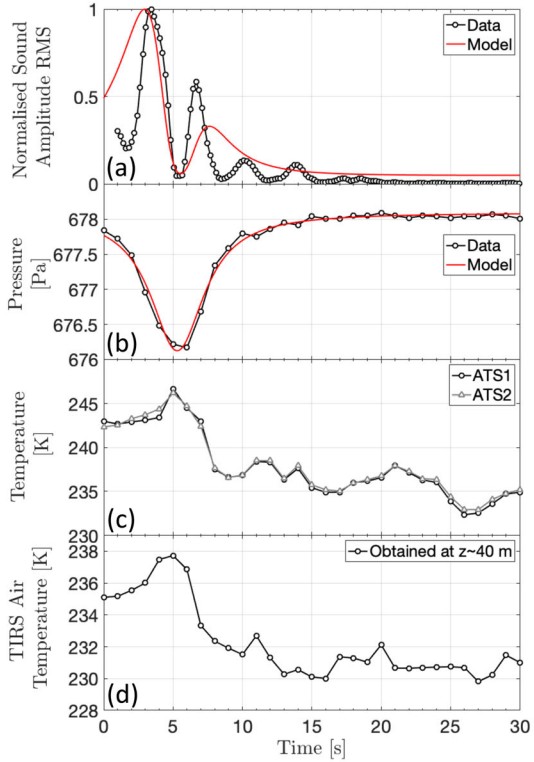

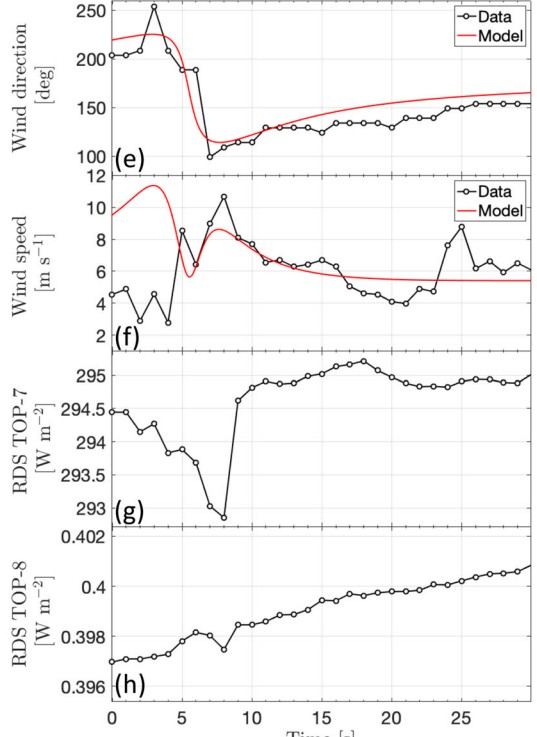

**Fig. 2 | Acoustic and multi-instrument recording of the direct dust devil encounter.** Data from the SuperCam microphone and the Mars Environmental Dynamics Analyzer (MEDA) instrument during the dust devil encounter. **a** The normalised microphone sound amplitude Root-Mean-Square (RMS) in the 20–60 Hz bandwidth calculated in 2-s windows, **b** the pressure data as measured by the MEDA barometer, **c** the atmospheric temperature at height ($z$) = 1.45 m as measured by the MEDA air temperature sensors (ATS) sensors, **d** the atmospheric temperature at the height of around 40 m as measured by the MEDA Thermal Infrared Sensor (TIRS), **e** the wind direction and **f** the wind speed as measured by the MEDA wind sensors, **g** the MEDA Radiation and Dust Sensor (RDS) TOP-7 sensor measurements, and **h** the MEDA RDS TOP-8 sensor measurements. In each of the panels, the data are shown in black and the modeling results are in red (see "Generating synthetic vortex data"), assuming the vortex parameters provided in Table 1. The time is indicated from the start of the SuperCam microphone recording. Source data are provided as a Source Data file.

proposed[20–24] but are still unverified on Mars[16]. A key measurement for testing existing theories, and thus correctly representing grain lifting in Martian atmospheric models, is the particle flux. Observing changing aeolian features on the Martian surface can provide grain flux estimates[25–27]. However, wind-blown grain fluxes have not yet been directly measured on Mars[15].

Convective vortices are typically identified using a barometer and manifest themselves as a sharp dip in the local pressure time series. Typical pressure amplitudes on Mars range from 0.3 to a few Pa, with durations from a few seconds to tens of seconds, but more extreme events with amplitudes up to[14] 10 Pa, or durations larger than[1] 250 s have also been observed. Vortices also generate other observables such as temperature increases, short duration changes in the wind speed and direction[28,29], magnetic signals[30], infrasounds[31], and even ground deformation signals[32,33]. The SuperCam microphone[34,35] onboard the Perseverance rover records air pressure fluctuations during 167 s periods with a sampling rate of 25,000 samples per second at the height of about 2.1 m above the Martian surface. As expected from pre-flight test campaigns[36] the intensity of the microphone signal in the 20 Hz to 1 kHz bandwidth has been shown to be strongly correlated with the wind speed[37]. As a result, the microphone can be considered as a high-frequency wind speed sensor capable of probing pressure variations associated with very short-term wind speed fluctuations[38] such as those produced by convective vortices. Convective vortices also generate acoustic waves in the infrasound band (<20 Hz)[31,39,40], but these signals are outside the frequency range of the SuperCam Microphone.

In this work, we describe the direct encounter of a dust-laden vortex with the Perseverance rover on September 27, 2021 (Perseverance sol 215, with areocentric solar longitude, Ls = 105°) during which the SuperCam microphone recorded the sound of the dust devil. First, we consider how likely it is to record such an encounter with the SuperCam microphone then we present the simultaneous measurements made by the microphone, the rover's Navigation Camera (Navcam) and the Mars Environmental Dynamics Analyzer (MEDA) instrument sensors. We determine the dust devil parameters and trajectory using this multi-instrument data set combined with vortex modeling. Finally, we describe the acoustic signals of grain impacts that were also recorded during the vortex encounter and we analyze these to provide quantitative information about the number density of particles in the vortex.

## Results
### A chance encounter

The MEDA instrument Radiation and Dust Sensor[41] (RDS) includes seven lateral-viewing sensors orientated at different azimuth angles (LAT detectors), and eight top detectors that view the zenith direction with different spectral filters ranging from the UV to near-IR wavelengths (TOP detectors). A systematic analysis of close encounters of dust-loaded pressure drops visible in the RDS signals resulted in the detection of 91 dust devils over the first 216 sols of the mission[7]. The dusty vortex observed at 11:02 local true solar time (LTST) on September 27, 2021 (Perseverance sol 215, with areocentric solar longitude, Ls = 105°) by the SuperCam microphone is a typical event in terms of its properties observed with the MEDA instrument (the

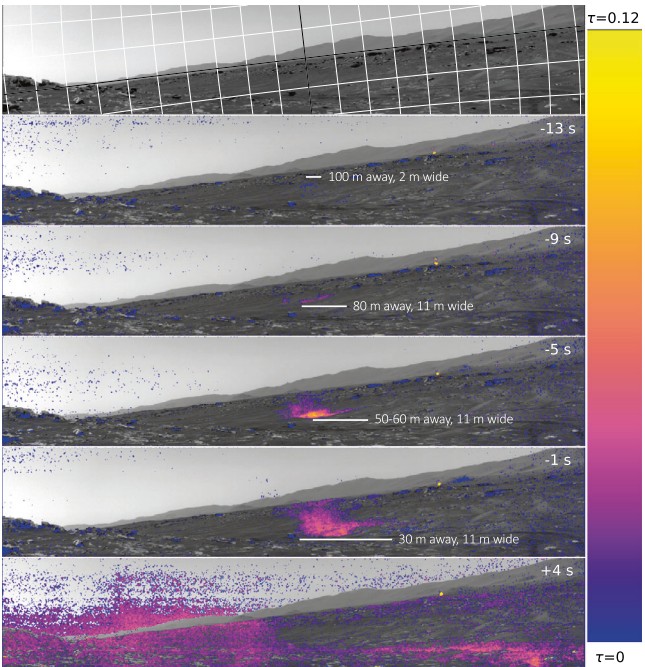

**Fig. 3 | The rover's Navigation Camera (Navcam) observations of the direct dust devil encounter.** The Navcam scene (the average background image, top panel) and each dust devil image processed to dust amount (five lower panels), masking out only the truly indeterminate areas (grey). The colour scale ranges from optical depth, $\tau = 0$ at the bottom (blue) to $\tau = 0.12$ at the top (yellow), and is linear in between. The areas with low signal-to-noise ratio are masked out, and the images also show non-random noise (banding from instrument electronics during read-out). The grid on the scene image (top panel) is $5°$ in local level azimuth and elevation; the darker grey contours are Elevation=0° and Azimuth=165°. The first dust devil image was taken at the rover spacecraft clock (SCLK) time of 686020326 s. The time indicated in the top right corners of the images is with respect to the start of the SuperCam microphone recording.

intensity of the pressure drop and the amount of dust are both close to the median of the distribution of events). A comparison with the cadence of MEDA[42] observations and the vortex detections[7] during that period suggests that a single microphone recording of 167 s obtained with LTST in the range of 11:00–16:00 has a 0.4–0.6% probability of detecting a comparable event to the one described here. However, there are 5062 s of joint SuperCam microphone–MEDA barometer atmospheric recordings with LTST in the range of 11:00–16:00 during the first year of the M2020 mission (sols 1-358). A Monte-Carlo simulation of a large survey of 5062 s of recordings over a period of time of similar activity to the one described in ref. 7 would result in a probability of 11–16% to find at least one similar event, making our detection a chance encounter. However, the fact that this dust devil encounter was also observed simultaneously by so many of the MEDA sensors and by the Navigation Camera (Navcam)[43] makes it an entirely unique event.

## Multisensorial recordings of a dust devil encounter

The 167 s SuperCam microphone acoustic recording on Perseverance sol 215 shows two 2–3 s periods of low-frequency signal content (20–60 Hz) close to the start of the recording, with an ~3 s very quiet period in between and after them (Fig. 1). Such low-frequency content is typical of microphone signals associated with wind gusts[37,38]. The Root-Mean-Square (RMS) of the sound amplitude (calculated in 2 s windows—see "Microphone data and data processing") has a similar double-lobed structure as predicted for very close encounter vortex signals recorded on the microphone[44], and is similar to acoustic data recorded on Earth during terrestrial vortex encounters (see "Acoustic

recordings of terrestrial vortices"). The two peaks correspond to the stronger winds at the vortex walls resulting in a larger amplitude microphone signal, while between the peaks the rover is in the calm eye of the vortex. The observed asymmetry in the RMS is indicative of a small offset between the background wind direction and the dust devil trajectory. The wind gusts are coincident with a pressure drop measured to be 2 Pa by fitting the barometer data to a standard vortex pressure model[4] (see "Wind speed, direction, and pressure data"), and a sharp change in wind direction as recorded by the MEDA wind sensors. This confirms that it is indeed a convective vortex (Fig. 2). MEDA's RDS measurements and optical images taken with the Navcam indicate that the vortex is lightly loaded with dust, producing changes in the irradiances measured by RDS of between 0.1 and 1% (Fig. 3 and see "Radiation and Dust Sensor analysis"). This places this vortex in the lower third quartile of the dust devils detected with RDS, with respect to irradiance changes. The air temperature sensors[42] (ATS) located at the mast of the rover indicate that there is 2–3 K of near-surface air temperature heating coincident with the vortex-induced pressure minimum (Fig. 2 and see "Temperature data during the vortex encounter"). However, this temperature excursion is similar in amplitude to the regular fluctuations in the temperature data at that altitude level. The Thermal Infrared Sensor (TIRS) measuring air temperatures at the height of approximately 40 m showed a similar response to the air temperature sensors, with a temperature peak above the noise level at the time of the vortex passage (Fig. 2 and see "Temperature data during the vortex encounter"). At the time of this dust devil encounter, the ground temperature in the field of view of MEDA was 257 K and the air temperature was 238 K at the height of MEDA (around 1.45 m). The atmospheric density at the height of the atmospheric temperature sensors was 0.0149 kg/m³, while the thermal inertia and the broadband albedo (0.3–3 μm) in the vicinity of 11:00 LMST were $425 \pm 25$ SI units and $0.14 \pm 0.01$, respectively[45] (see "Thermal inertia and albedo context information"). Assuming cyclostrophic balance (i.e., the pressure gradient provides the force needed for the centripetal acceleration of the rotating air), the vortex wind $V_T$ can be approximated by

$$V_T = (\alpha \Delta P / \rho)^{0.5} \qquad (1)$$

where $\Delta P$ is the core pressure drop, $\rho$ is the air density and $\alpha$ is a parameter that can range from 0.5 to 1.0, where 0.5 indicates a fully cyclostrophic vortex[8,46] ($\alpha$ is estimated to be 0.52 for this particular vortex, see "Monte-Carlo simulations for vortex parameter determination"). The peak vortex wind speed for this event is, therefore, likely to be closer to 8 m s⁻¹. When combining the background wind (measured by the MEDA wind sensor to be 5.7 m s⁻¹, see "Wind speed, direction, and pressure data") and the local vortex winds, this leads to a theoretical peak wind velocity of ~11 m s⁻¹ (see "Generating synthetic vortex data"). This is compatible with the maximum wind speed measured by the MEDA wind sensors, which is 10.7 m s⁻¹, and the maximum wind speed estimated from the microphone data, which is close to 10.5 m s⁻¹ (see "Wind speed estimation with the microphone").

## Determination of the dust devil parameters and trajectory

The dust devil can also be seen in Navcam images taken as part of Dust Devil Movie (Fig. 3). Visible 9 s before the beginning of the microphone recording, the dust devil initially appears as a columnar dust cloud moving towards the rover then as a dust devil. Then, around 4 s into the microphone recording, the dust fills the Navcam field of view as the dust devil completely envelopes the rover (Fig. 3). The dust devil is seen to be coming from an azimuth (Az) of around 164° (from approximately SSE) in the Navcam images (where Az = 0° is North and increases clockwise), initially as a 2 m dust cloud 110 m away, and then as an 11 m diameter cloud in subsequent images (Fig. 3 and see "Navcam image

**Table 1 | Derived vortex parameters for the dust devil at 11.02 LTST on September 27, 2021**

| Parameter | Value |
|---|---|
| Diameter | 25.0 +/– 1.56 m |
| Core pressure drop | 1.97 +/– 0.05 Pa |
| Trajectory | From 164.3 +/–1.4° |
| Translation speed | 5.3 +/– 0.3 m s⁻¹ |
| Rotation direction | Clockwise |
| Closest approach distance | 0.17 +/– 0.22 m (direct hit) |
| Vortex height | >118 m |

(Perseverance sol 215; Ls = 105°) The uncertainties represent the standard deviation of the best four solutions of an extensive Monte-Carlo parameter fitting study (see "Monte-Carlo simulations for vortex parameter determination").

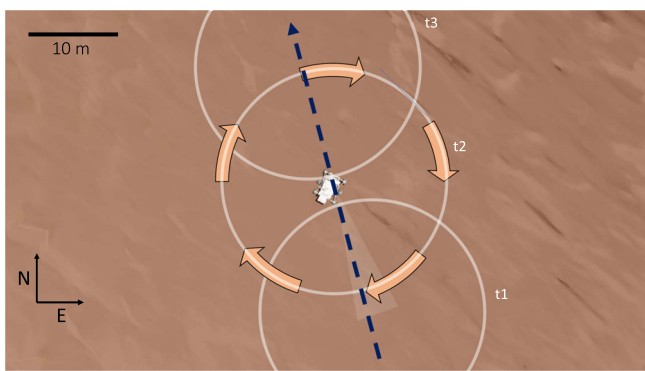

**Fig. 4 | Dust devil trajectory.** The dust devil trajectory direction (blue dashed arrow) and size (white circles) with respect to the Perseverance rover and the local terrain at the time of the encounter. The dust devil has a trajectory coming from Azimuth (Az) = 164°, and the vortex diameter is estimated from combined data and modeling to be 25 m. The rover Navigation camera (Navcam) field of view is indicated by the pale triangle. At $t_1$ Perseverance is in the leading vortex wall, at $t_2$ Perseverance is inside the eye of the vortex, and at $t_3$ Perseverance is in the tailing vortex wall. The vortex and the rover are drawn to scale. The orange arrows indicate the clockwise rotational direction of the vortex winds.

interpretation"). One would expect the dust to be primarily in the high wind area of the vortex wall, however, it would appear from the images that the majority of the dust is in the center of the vortex (see "Navcam image interpretation"). With a sampling rate of up to two samples per second, the wind sensor is not well suited to resolve the rapid wind fluctuations associated with this convective vortex. However, the high microphone sampling rate (25 kHz) allows the faster dynamics to be studied. The drop in the wind speed between two gusts, coincident with the pressure minimum, indicates that this dust devil passed directly over Perseverance. The initial wind gust visible in the microphone data corresponds to the leading vortex wall, and the second wind gust to the trailing vortex wall. In between these gusts, Perseverance is in the calm eye of the vortex with a wind speed <2 m s⁻¹ (the limit of sensitivity of the microphone to wind speed[38], Supplementary Fig. 7). In the final Navcam image, the amount of dust is close to half that in the penultimate image thus providing further evidence that the image is taken from the middle of the dust devil. Using Monte-Carlo modeling of the pressure and wind data assuming a drifting vortex model[47], the dust devil parameters have been well-constrained (Table 1 and "Monte-Carlo simulations for vortex parameter determination"). These parameters are also consistent with the thermal sensor and RDS data ("Radiation and Dust Sensor analysis" and "Temperature data during the vortex encounter"). The dust devil spends 12–13 s in the RDS TOP-8[41] sensor (±15° field of view at zenith) which requires, for a translation speed of 5.3 m s⁻¹, a minimum vortex height of about 118 m (see "Radiation and Dust Sensor analysis"). Although dust devils follow the approximate ambient wind direction[48] they do not always translate with exactly the same speed and direction as the background wind measured at the surface. Indeed, vortices may actually translate with boundary layer winds that can be up to 2 times faster than surface winds[49]. Assuming the derived dust devil parameters (Table 1), and a background wind speed and direction of 5.7 m s⁻¹ and 181° (from S to N), respectively (see "Wind speed, direction, and pressure data"), synthetic sensor data are generated ("Generating synthetic vortex data") and compared directly with the sensor measurements (Fig. 2). The synthetic pressure (Fig. 2b) and wind direction models (Fig. 2e) match the observations well. The asymmetry observed in the sound amplitude RMS due to the offset between the background wind direction (coming from Az = 181°) and the dust devil trajectory (coming from Az = 164°) is captured by the synthetic model (Fig. 2a). The synthetic wind speed model also corresponds well to the measured data from the point of closest approach (the pressure minimum), but there is a discrepancy between wind speed synthetics and data in the first part of the encounter (Fig. 2f). This may be due to the low sampling rate of the MEDA wind sensor and highlights that acoustic data are ideal for resolving the rapid wind structure of

vortices and complementing measurements with the MEDA wind sensors, which have a response time on the order of seconds. The dust devil size and trajectory with respect to the Perseverance rover are shown in Fig. 4.

## Quantifying the vortex grain loading

In addition to the low-frequency wind signal, the acoustic recording of the dust devil contains high-frequency impulse signals (2–4 kHz) associated with grain impacts (Figs. 1d and 5). This high-frequency saltation "hiss" is very similar to reported terrestrial saltation signals[50] that are the result of direct impacts on a microphone diaphragm. For this SuperCam microphone recording, it is not possible to know exactly where the grains are impacting; the microphone signals we observe may be purely acoustic signals, or may be the result of a mechanical transmission of the impact energy through the structure to the microphone. However, given the small impact energies of lofted grains and the large acoustic attenuation of the Martian atmosphere[37,51–53], it is likely that the impacts heard in the sound recording are local, possibly on the microphone (finger or tip[34]) or close to it on the cover of the SuperCam Mast Unit[35] at the top of the rover mast. Using the algorithm presented in ref. 50 for terrestrial acoustic sensors, individual grain impacts that occurred during the vortex encounter were detected (see "Signal processing to obtain grain impact frequency"). In total, 308 grain impacts were identified during the vortex encounter, distributed between three bursts of duration 0.8, 1.2, and 1.4 s. Within these three bursts the rate of detected impacts was 77 impacts/s, 144 impacts/s, and 78 impacts/s, respectively with a mean grain impact frequency of 60 impacts/s across the entire event (Fig. 4). This number is consistent with the impact frequency measured on Earth[50], which was between 60 and 90 impacts/s. Having three bursts of grain impacts is surprising: the grains would be expected to be within the vortex walls and, therefore, impacts should occur predominantly in the regions of peak vortex winds. This is the case for the first and third bursts, which correspond to the leading and trailing vortex walls. The second peak occurs very close to the vortex center and may simply be due to additional sand picked up by the vortex just before the encounter with the rover, or this may be further evidence of the unusual dust cloud observed by the images in the center of the vortex. Assuming that the dust devil's translation speed is 5.3 m s⁻¹, this leads to a maximum value of 27 grain impacts/m within the vortex for the largest observed impact frequency. This low grain content is consistent with the RDS measurements and the optical

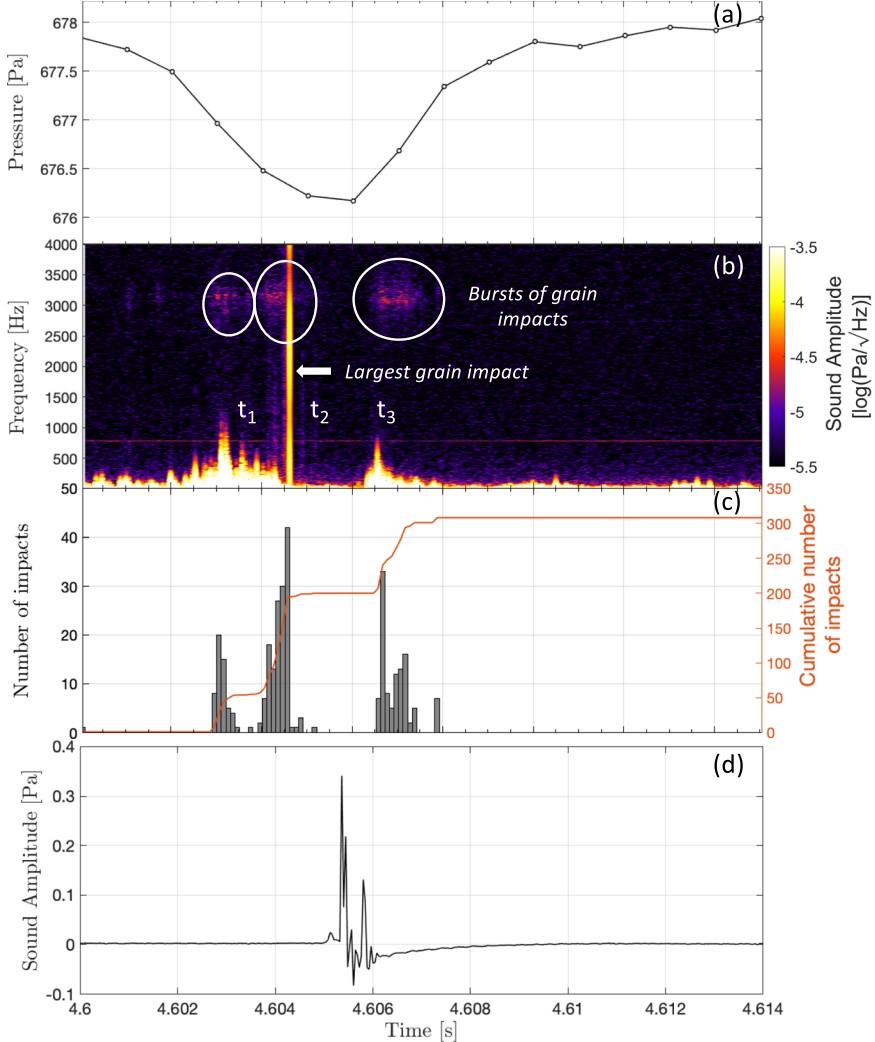

**Fig. 5 | Identification and quantification of grain impacts. a** The pressure data during the dust devil encounter, **b** the spectrogram of the microphone sound amplitude calculated using a window size of 0.2 seconds in order to resolve the fine details in the acoustic data, **c** a histogram (gray) and cumulative histogram (orange) of the grain impacts as a function of time and, **d** the microphone sound amplitude time series zoomed in around 4.6 s to show the loud, very broadband, signal. Three bursts of grain impacts can be observed in both the spectrogram (**b**) and histogram (**c**). At $t_1$ Perseverance is in the leading vortex wall, at $t_2$ Perseverance is inside the eye of the vortex, and at $t_3$ Perseverance is in the tailing vortex wall (see Fig. 4). Source data are provided as a Source Data file.

depth estimates from the images, both of which observed a very low dust content within this vortex (Fig. 3 and "Navcam image interpretation"). Finally, there is also a loud, and a very broadband signal (a "bang") that occurred 4.6 s after the start of the audio recording (Fig. 5), just after the leading vortex wall had passed the microphone. The notch in signal amplitude close to 6 kHz (Fig. 1d), which is a result of destructive interference of the acoustic wave due to echoes from the base of the microphone finger[37], confirms the acoustic origin of this large signal. This indicates that the sound has been reflected from the front surface of the SuperCam Mast Unit[35], in order for this interference to occur. Given that this loud signal occurs within the second burst of grain impacts, and has a typical impulse response[50] (Fig. 5d), we suggest that it is caused by a particularly close impact, or by the impact of a much larger particle.

## Discussion

This chance dust devil encounter demonstrates that acoustic data are ideal for resolving the wind structure of vortices and complementing measurements with thermal anemometers, which have a response time on the order of seconds. Acoustic recordings can also be used to determine whether a vortex encounter is a direct hit (double-lobed

acoustic signal), even in the absence of other wind, temperature or dust sensors. An important result from these data is the detection of grain impacts which opens up the opportunity to directly observe the wind conditions under which energetic grain motion occurs. This is vital for understanding Martian saltation and dust lifting, which are the dominant causes of surface change and (in the case of dust) climate variability in the present and recent past of Mars. The microphone observations also provide the possibility of quantifying the wind-blown grain fluxes on Mars. Such measurements will lead to an improved understanding of surface lifting and atmospheric transport, information that is key for accurate simulation of the dust cycle and prediction of dust storms in Mars climate models. Future laboratory experiments investigating the relationship between the kinetic energy of a grain impact and the microphone output would be useful to determine the mass flux of grains using acoustic data, in addition to quantifying the distance at which the grain impact would be heard. Grain impacts are implicated in the degradation of hardware on the surface of Mars[54], thus measuring the probability or rate of grain impacts may help with future instrumentation designs or in planning operations. Such investigations would also permit on-board data compression schemes to be formulated to more efficiently extract

information on aeolian processes from microphone data, which as presently implemented on Perseverance requires substantial data volumes to be downlinked (8 MB per 167 s recording). Additional microphone recordings during the convective period on Mars may provide the opportunity to observe more dust devils as the Perseverance mission continues. This would allow comparative studies to be performed of the dust-lifting capabilities of different vortices and at different geographic sites.

## Methods

Here, we describe the microphone data and applied processing, examples of terrestrial acoustic recordings of vortices, the Mars Environmental Dynamics Analyzer (MEDA) wind speed, wind direction, and pressure data, and also the pressure drop fitting, MEDA Radiation and Dust Sensor analysis, the MEDA temperature data and inferred thermal inertia during the vortex encounter, thermal inertia context information, wind speed estimations with the SuperCam microphone, the Navcam image interpretation, Monte-Carlo simulations for vortex parameter determination, how the synthetic vortex data are generated, the signal processing applied to determine the grain impact frequency from the acoustic data and some considerations about the local nature of the grain impacts based on the acoustic attenuation of the Martian atmosphere.

### Microphone data and data processing

SuperCam's microphone records air pressure fluctuations from 20 Hz to 12.5 kHz at a 25 kHz sampling frequency, and up to 50 kHz when the 100 kHz sampling mode is used. The analog signal from the microphone, ranging from 0 to 5 V, is digitized (12-bit depth) using one of 4 electronic gains to boost the sensitivity from 0.6 to 21 V/Pa and the resolution from 2 to 0.06 mPa. For atmospheric recordings such as the one on Perseverance sol 215, the highest gain level is used (21 V/Pa), corresponding to an amplification factor of 972. The microphone's electronic response function is used to correct raw spectra below 100 Hz. The spectrogram of the microphone sound pressure level is calculated using a window size of 0.2 s in order to resolve the fine details in the acoustic data. For the large grain impact, there is a drop in signal amplitude close to 6 kHz (Fig. 1). This is a result of destructive interference of the acoustic wave due to echoes from the base of the microphone itself, as seen in the pre-flight microphone tests and confirmed previously on Mars[37]. This confirms the acoustic origin of the impact signal. Also visible in the spectrogram is the acoustic tone at around 760 Hz produced by a pump on the rover.

### Acoustic recordings of terrestrial vortices

A terrestrial sound recording of a convective vortex can be seen in Supplementary Fig. 1. The microphone (not the SuperCam Microphone) signal rises appreciably during the vortex encounter, and the double peak indicates that the calm "eye" of the vortex passed very close to the sensor, with the stronger winds at the vortex "wall" giving a stronger microphone output. These data were obtained during a field experiment at Goldstone Dry Lake, California USA in June 2014[44].

### Wind speed, direction, and pressure data

The wind and pressure data are recorded from the Mars Environmental Dynamics Analyzer (MEDA) instrument[42]. These data are acquired at 1 sample per second during this event. Wind speed and direction are independently acquired from two separate booms, both at the height of 1.5 m above the ground and separated by 120° (termed boom 1 and boom 2). Data from one boom is generally superior to the other, for a given wind direction, depending on the rover elements obstructing the flow to each boom. The accuracy of the wind speed measurement is $\pm 1$ m s$^{-1}$ (up to 10 m s$^{-1}$) and 10% of the wind speed over 10 m s$^{-1}$ and the resolution is $\pm 0.5$ m s$^{-1}$ (up to 10 m s$^{-1}$) and 10% of the wind speed over 10 m s$^{-1}$. The wind direction is resolved with an accuracy of $\pm 15°$. The

barometer is at the height of 1 m above the ground, and has an accuracy < $\pm 5$ Pa and a resolution of $\pm 0.12$ Pa. The observed pressure drop amplitude is estimated to be 2.00 Pa by fitting the pressure data to a standard vortex pressure profile[4] (Supplementary Fig. 2a). The goodness of fit[33] between the model and data is >0.9. The encounter duration, assumed to be twice the Full-Width Half Maximum (FWHM) of the encounter duration, is 9.35 s. The average wind speed and direction measured by the MEDA wind sensor in the 147 s period following the vortex encounter (the remaining duration of the microphone recording) is found to be 5.7 m s$^{-1}$ and 181°, respectively (Supplementary Fig. 2b). This is assumed to be the background wind speed and direction at the time of the vortex encounter.

### Radiation and dust sensor analysis

The Radiation and Dust Sensor[41] (RDS) includes seven lateral-viewing sensors orientated at different azimuth angles (LAT detectors), and eight top detectors that view the zenith direction with different spectral filters ranging from the UV to near-IR wavelengths (TOP detectors). The RDS-LAT (field of view $\pm 5°$ pointing sideways, 20° above the horizon) and RDS -TOP (field of view $\pm 15°$ at zenith, except for TOP-7 with $\pm 90°$) sensor observations show variations produced by the presence of the dust devil (Supplementary Fig. 4). For LAT sensors, only LAT-7 and LAT-8 (both looking in the direction of the approaching dust devil—see Supplementary Fig. 3a) present variations greater than 3-$\sigma$ (threshold defining a dust devil detection). The LAT-6 sensor did not detect the dust devil despite being pointed toward the dust devil azimuth direction. This is because the dust devil was not tall enough (likely not well constructed or transporting a low concentration of dust) to be detected by the LAT-6 sensor, and its trajectory was quasi-parallel to the sensor FoV (Supplementary Fig. 3b). It is only in the last Navcam frame (+8 s with respect to the microphone recording start) that the dust devil is tall enough to cover the LAT-7, LAT-8, and TOP channels. Based on the order of these variations and by comparing with previous detections, we infer a low dust devil opacity (changes in the irradiances measured by RDS lower than 1%, corresponding to an event in the lower third quartile of the dust devils detected with RDS). The sequence of detections and orientations is consistent with an incoming direction a little bit closer to South–South–East than purely South. The TOP-7 sensor indicates a sun-direct-light blocking at 11:02:36 LTST (see Supplementary Fig. 4) for which the sun azimuth angle, and therefore that of the dust devil as well, was 64.59°. From the sun-direct-light blocking duration $\Delta t$ and the dust devil's translation speed (v) and trajectory, the diameter can be estimated by:

$$Diameter = v \times \Delta t \times \sin(\delta) \qquad (2)$$

where $\delta$ represents the angle between the dust devil trajectory and sun azimuth. From $\Delta t$ in Supplementary Fig. 4 and the information given in Table 1, we estimate from RDS observations a dust devil diameter of $28.8 \pm 4.3$ m (consistent with the Monte-Carlo-derived diameter estimate provided in Table 1). Information on the dust devil height can also be derived from the detection carried out by the TOP-8 sensor ($\pm 15°$ at zenith). The bottom panel of Supplementary Fig. 4 shows the dust devil crossed the TOP-8 field of view for about 12–13 s, which requires, for a translational speed of 5.3 m s$^{-1}$, a minimum height of about 118 m.

### Temperature data during the vortex encounter

Data from the five Atmospheric Temperature Sensors[42] (ATS) and from the Thermal Infrared Sensor (TIRS) ground and 40 m temperature retrievals are provided in Supplementary Fig. 5. ATS1-3 are mounted at 1.45 m on the rover's remote sensing mast, while ATS4 and 5 are mounted at 0.85 m on either side of the front of the rover. ATS1 and ATS2 have good measuring conditions and are the best proxy for the

near-surface air temperature at the time of this vortex encounter. ATS4 does not show fluctuations because it is "sheltered" from the environment at this time and records the temperature within the rover's thermal boundary layer. This is also the case for ATS3, but to a lesser extent, which records a constant temperature over most of this period. The abrupt and asymmetric change in ATS5 is due to the wind direction change, as the temperature sensor is cooled when the sensor points toward the wind direction. This effect is likely not related directly to the vortex thermodynamics but the wind direction change is a consequence of the vortex winds. There are no vortex features in the ground temperature data provided by TIRS. The TIRS sensor measuring air temperatures at a height ($z$) of approximately 40 m showed a similar response to ATS1 and 2, with a temperature peak well above the noise level at the time of the vortex passage. Winds during the vortex encounter were such that heat from the rover's Radio-isotope Thermonuclear Generator (located at the rear of the rover), should have been advected away from the vortex, allowing a good measurement of temperatures. From the combination of ATS measurements, we found a temperature change $\Delta T$ = +1.0 K at the core of the vortex at $z$ = 0.85 m and $\Delta T$ = +5.0 K at the core of the vortex at $z$ = 1.45 m (although this is well inside the typical temperature oscillations at that time), and from TIRS we found a clear peak of $\Delta T$ = +5.0 K at the height of ~40 m.

## Thermal inertia and albedo context information

Due to conjunction, the rover was stationary during a period of 26 sols from sol 211 to 237. During the entirety of this period, the MEDA-derived thermal inertia was $425 \pm 25$ SI units, while the broadband albedo in the vicinity of 11:00 LTST was $0.14 \pm 0.01$. This value of thermal inertia is among the highest observed by Perseverance, which ranged from 180 to 600 tiu over the first 350 sols of the mission, with a mean value of around 350 tiu[45]. The value of the albedo at around 11:00 LTST was remarkably constant from sol 211 to 237. Contrary to some other close dust devil encounters, the albedo did not change abruptly during the passage of the vortex on sol 215[55]. This could be due to the geometry of the encounter, as the vortex approached the rover with an azimuth angle of 164°, while the TIRS' field of view was located at an azimuth angle of 102.6°, and covers an ellipsoid area of only 3–4 m² of the surface about 3.75 m from the rover RTG (to avoid thermal contamination), or because the vortex had stopped raising dust by the time it passed over the TIRS field of view. For context, Supplementary Fig. 6 shows a mosaic of the surrounding area and the approximate field of view of TIRS used to determine the thermal inertia and albedo.

## Wind speed estimation with the microphone

The microphone records pressure fluctuations above 20 Hz. The wind-induced noise observed by the (unshielded) microphone depends on the pressure fluctuations in the wind flow (i.e., turbulence) and also from the interaction of the microphone with the wind flow[56–59]. In each case the wind noise is correlated to the wind speed. The observed sensitivity of the microphone to the wind speed can, therefore, be considered as an in situ calibration for the microphone to provide an estimation of the wind speed. To that end, Supplementary Fig. 7 demonstrates the relationship between the wind speed and the signal power of the SuperCam microphone observed on Mars. For wind speeds above 2 m s⁻¹ the relationship can be approximated by a fourth-order power law. To that end, we obtain a simple estimate of the wind speed from the microphone by taking the fourth root of the RMS envelope in the 20–60 Hz bandwidth and normalizing the signal compared to the model. This estimation is most applicable to high and fast varying wind speeds such as associated with vortex encounters, as analyzed in this work. A detailed treatment of the sensitivity of the microphone to the wind and other atmospheric data is provided elsewhere[38].

## Navcam image interpretation

The rover orientation and location on Perseverance sol 215 are shown in Supplementary Figs. 8 and 9, respectively. The Navcam images place the dust devil as coming from an azimuth of approximately 163° (162° from the Navcam image, 164° from a comparison with orbital data using the Multi Mission Geographic Information System—MMGIS), initially as a 2 m dust cloud 110 m away, and then as an 11 m diameter cloud in subsequent images taken over the following 21 s (Fig. 3 and Supplementary Figs. 10–14), suggesting that either the vortex was still growing, or dust was being initially entrained into the vortex, when it was first observed. The diameters are based on hand drawings of the edge of the dust cloud. The distance in the 30-m image is well-constrained, since it is just about to cross rover tracks, which are well-defined in length. The diameter estimates of the dust clouds remain subjective and the uncertainty is probably >10% in any individual image. However, the 11 m diameter was derived from three separate and consecutive images. The diameter of the vortex is estimated to be 24–28 m, translates at 4.0–5.7 m s⁻¹, and has peak rotational winds of 10–12 m s⁻¹ in a clockwise direction (consistent with the Monte-Carlo-derived diameter estimate provided in Table 1). The rotational period is $T$ = 6.3–8.8 s. The image spacing was about 4 s between the first four images and almost 5 s to the last i.e., 0.45–0.63 $T$. If $T$ is around 8 s, the dust cloud would do one rotation each second image; for anything else, it would progress around. If the vortex center is left or right of the dust cloud in image 0, then it must move 24–28 m side to side; this is not observed. If the vortex center is between the rover and cloud (or vice versa), then the dust cloud appears to move in a straight line by coincidence as it traced out its cycloidal path. If $T$ were not 2× image spacing, the apparent motion would have to be more complex, which was not observed. Therefore, the imaged dust is unlikely to represent a dusty area in a larger, otherwise clear vortex. The images indicate that this dust devil is unusually amorphous, and the visible dust cloud appears to be moving in a linear fashion, at the vortex center. Dust optical depth was derived by computing a mean-frame image (including images not shown here) and then differences of each image from the mean. The radiative transfer equation was used to derive optical depth following ref. 60. The maximum derived optical depth is 0.12 (Fig. 3).

## Generating synthetic vortex data

The pressure is modeled using the pressure profile presented in ref. 4 i.e., the observed pressure drop $\Delta P_{obs}$ is given by

$$\Delta P_{obs}(r) = -\Delta P / (1 + r^2) \tag{3}$$

where $\Delta P$ is the core pressure drop and $r$ is the distance from the vortex center. The tangential vortex wind velocity ($V_T$) is calculated from cyclostrophic balance (Eq. (1)). The observed wind vector ($V$) during the vortex encounter is a combination of the background wind vector ($U$) and the local vortex wind vector at a distance $r$ from the vortex center as follows:

$$V = 2rV_T / (1 + r^2). \tag{4}$$

The background and local vortex winds are combined following the methodology presented in ref. 47 i.e., the observed wind coming from the East ($W_E$) and North ($W_N$) directions is given by

$$W_E = -U \sin\Omega + V \cos\theta \tag{5}$$

and

$$W_N = -U \cos\Omega + V \sin\theta, \tag{6}$$

respectively, where $\Omega$ indicates the azimuth of the ambient wind and $\theta$ is the observed azimuth of the vortex. The wind and pressure synthetic data are shown in Supplementary Fig. 15. The synthetic microphone RMS envelope is estimated from the synthetic wind speed assuming that the microphone RMS is proportional to the wind speed to the fourth power ("Wind speed estimation with the microphone"). For a vortex advected in the direction of the ambient wind, the resulting observed winds speed is double-lobed and symmetrical (Supplementary Fig. 16). However, when the ambient wind direction and the vortex propagation direction are not aligned, this leads to an asymmetrical double lobe, as observed in the microphone sound amplitude RMS (Fig. 2). The resulting synthetic model and vortex trajectory are shown in Supplementary Fig. 15. Also shown, for comparison, are the wind speed synthetics if the vortex trajectory and wind directions were to be perfectly aligned, or if the vortex wall rather than the vortex center passed over the rover (Supplementary Fig. 16).

### Monte-Carlo simulations for vortex parameter determination
We use the drifting vortex model following ref. 47 (described in "Generating synthetic vortex data") and the tangential vortex wind velocity is calculated from cyclostrophic balance (Eq. (1)). We also consider the pressure and air temperature to give a good evaluation of the air density. To find a solution we use a Monte-Carlo exploration of the wide space of parameters in different steps. Step 1 is a blind evaluation of 20,000 models covering a wide range of parameters and looking for the ten best solutions. In Step 2, we run another 20,000 simulations but in a restricted space of parameters centered around the solutions found in the step before. Then, the third and final step involves another 20,000 simulations centered in the space of parameters found in Step 2. The results of the final step and the trajectories of the best four solutions are shown in Supplementary Figs. 17 and 18.

### Signal processing to obtain grain impact frequency
An established method[50] is used to count the grain impacts. The first step is to filter the acoustic signal in the relevant frequency band, which is around 8 kHz on Earth but between 2 and 4 kHz in the case of this dust devil. Further events will have to be analyzed to determine whether the different frequency band is recurrent. Then, the signal is smoothed by subtracting its three-point running mean and all values smaller than zero are set to zero. The mean of the time series is removed and all values smaller than a certain threshold are removed. The determination of this threshold is crucial as choosing one too low would lead to removing peaks produced by grain impacts, while choosing one too high would lead to counting as grain impacts peaks that are merely noise. The threshold proposed in ref. 50, four times the standard deviation of the time series, was found to be optimal because it leads to the detection of grain impacts at the exact same times when they are visible on the spectrogram. The last step is to detect the peaks (local maximum) of the processed signal. Peaks are defined as all points that are larger than the two neighboring samples. Supplementary Fig. 19 illustrates this processing on a portion of the signal lasting 0.5 s.

### Acoustic attenuation on Mars
Given the large acoustic attenuation of the Martian atmosphere, in addition to the geometrical spreading of the acoustic wavefront, the sound amplitude decreases rapidly with distance. As the acoustic wave propagates, the sound pressure decreases as $1/r^*e^{-\alpha r}$, where $r$ is the distance between the acoustic source and the microphone and $\alpha$ is a frequency-dependent atmospheric attenuation coefficient estimated to be close to 0.1 in the 2–4 kHz bandwidth[37,51]. At a distance of 1 m from the source, the sound amplitude in the 2–4 kHz bandwidth (the frequency band containing the grain impacts sounds) is already ten times smaller than the source amplitude, and this decreases to 100

times smaller than the source amplitude at a distance of 6 m (Supplementary Fig. 20). For alternative attenuation models[52], this acoustic attenuation would be even stronger. If the microphone signal generated by the impacts is acoustic, this implies that the recorded grain impacts are very local (likely on the SuperCam instrument, and perhaps also the closest parts of the rover). However, without knowledge of the source amplitude, we cannot provide a definitive distance at which the acoustic signal of the impacts will be heard. If the smaller impact signals are due to mechanical transmission of energy through the structure to the microphone, the low speed and small expected mass of the lofted grains imply a small impact energy and thus also a local impact.

## Data availability
The Mars2020 data used in this study is publicly available at the Planetary Data System Geosciences Node: https://pds-geosciences.wustl.edu/missions/mars2020/. The acoustic and MEDA data used here can be found at https://doi.org/10.17189/1522646 and https://pds-atmospheres.nmsu.edu/PDS/data/PDS4/Mars2020/mars2020_meda/, respectively. The images used in this paper are available at https://mars.nasa.gov/mars2020/multimedia/raw-images/. Source data are provided with this paper.

## Code availability
The codes used to perform the drifting vortex analyses[8] (including those to analyze MEDA data) can be accessed here: https://doi.org/10.5281/zenodo.6958141.

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

## Acknowledgements

We are most grateful for the support of the Mars 2020 project team, including hardware and operation teams. This project was supported in the US by the NASA Mars Exploration Program, and in France by CNES. It is based on observations with SuperCam embarked on Perseverance (Mars2020). The research carried out at the Jet Propulsion Laboratory, California Institute of Technology, is under a contract with the National Aeronautics and Space Administration (80NM0018D0004). The JPL co-author (M.T.) acknowledges funding from NASA's Space Technology Mission Directorate and the Science Mission Directorate. A. V-R is supported by the Spanish State Research Agency (AEI) Project No. MDM-2017-0737 Unidad de Excelencia "María de Maeztu"- Centro de Astrobiología (INTA-CSIC), and by the Comunidad de Madrid Project S2018/NMT-4291 (TEC2-SPACE-CM). R.H. and A.S-L. were supported by Grant PID2019-109467GB-I00 funded by MCIN/AEI/10.13039/501100011033/ and by Grupos Gobierno Vasco IT1742-22. A.M. was supported by Grant PRE2020-092562 funded by MCIN/AEI/10.13039/501100011033 and by "ESF Investing in your future". R.L. acknowledges InSight PSP Grant 80NSSC18K1626 as well as the Mars 2020 project. B.C. is supported by the Director's Postdoctoral Fellowship from the Los Alamos National Laboratory, grant 20210960PRD3. JA.RM., M.M, J.T and J.G-E were supported by MCIN/AEI's Grant RTI2018-098728-B-C31.

## Author contributions

N.M. led the writing of the manuscript and the analysis of the data. A.S., M.G., R.H., M.L., D.T., V.A., R.L., B.C., A.M., A.S-L., A.V-R., T.B., D.B., X.J., and C.N. performed data processing and interpreted the data. M.T. was acting as the M2020 atmosphere working group representative during the Perseverance sol 215 operations. S.N., M.M, J.T., and J.G.-E. are part of the MEDA team and provided the calibrated wind data. A.C. and A.So. built and tested the SuperCam microphone, D.M. is the lead of Super-Cam's microphone, S.M. and R.W. are the leads of SuperCam investigation, JA.RM., M.T., and C.N. are leading the MEDA investigation and mission atmospheric working group. All other co-authors provided helpful comments and inputs to the manuscript.

## Competing interests

The authors declare no competing interests.
