## [Peer Review File · Nature Communications]

REVIEWER COMMENTS

Reviewer #1 (Remarks to the Author):

"The Sound of a Martian Dust Devil" by Murdoch et al., presents a novel investigation of a suite of measured characteristics of a dust-laden convective vortex impinging upon a craft on the surface of Mars, in this instance the Perseverance Rover. First-of-their-kind acoustic measurements of wind and sand grain impacts are described, resulting in derived vortex wind speed and grain impact quantification, which enhance interpretation of other more traditional near-surface atmosphere measurements and suggests that future measurements and analyses might offer the ability to quantify grain flux, which is an integral factor in establishing the martian dust cycle.

The manuscript presents a compelling description of and correlations between a number of environmental measurements diagnostic of the directly detected vortex characteristics (translation speed, vortex rotational wind speeds, atmospheric pressure deficit magnitude, gas temperature, atmosphere dust load) and derived vortex characteristics (diameter, height). Acoustic data provide novel sounds of a dust devil in the form of both lower frequency (10s of Hz) signals identified as diagnostic of wind signals, and higher frequency (few thousand Hz) measurements characteristic of grain impacts upon rover components.

It is exciting to see the capability of the suite of Perseverance instrumentation to provide such a thorough representation of conditions present with a martian dust devil, and very compelling to see the valuable investigative opportunities these data offer as presented in this manuscript.

I provide a few minor comments/thoughts about the manuscript's contents which the authors' might consider addressing to improve the experience for the reader.

1. The Perseverance RDS measurements are introduced (text line 85, 115, 151, ...) prior to the 'Radiative Dust Sensor' being described on text line 321; it would aid the reader less familiar with the MEDA instrument suite to introduce the Radiative Dust Sensor prior to or contemporaneously with the initial invocation of the RDS measurements

2. In regard to the prominent acoustic signal occurring at the 4.6 second moment, indicated to be the sound signal of the impact of a large sand grain: is there any laboratory work available to estimate the size/mass of the grain required to produce a signal of such magnitude? With this impact occurring when the rover was within the 'calm' vortex interior (within the radius of maximum

rotational winds), and the possible small magnitude wind speeds experienced there, is there some lower limit to gran mass than can be inferred (with appropriate assumptions such as the distance of the impact from the microphone)?

3. Section 6 of the METHODS section possibly suggests that time-of-sol thermal inertia values are relevant in addition to time-of-sol dependent albedo. It might help the reader if the presentation of the 425 +/- 25 SI unit value of the thermal inertia was indicated to be representative of the thermal inertia during the entirety of the 26 sols Perseverance was at this location, if this is a correct statement

4. There is redundancy of the information presented in Figures 1d, 5b, and Supplementary Figure 1 (which seems to be referenced as "Fig. 1" on text line 293; Figure 5 provides a helpful visual comparison between the measured vortex pressure signature, the derived acoustic spectrum and other characteristics of the acoustic environment; however, Supplementary Figure 1 does not add any new information compared to Figure 1 and the removal of Supplementary Figure 1 would not harm the manuscript's presentation. Additionally, Supplementary Figure 1 is not referenced within the main body of the manuscript nor in the METHODS Section. [And, Supplementary Figure 2 is not referenced within the manuscript main body or within the METHODS section.]

5. The graphed lines in the top four frames of Supplementary Figure 5 are very faint and could be difficult for the reader to see sufficiently

Reviewer #2 (Remarks to the Author):

This paper reports on the pressure signal recorded on the Mars Perseverance SuperCam microphone during the passage of a dust devil. Prior to this, only measurements of pressure fluctuations due to turbulence (wind noise) and signals associated with the Mars mission (anthropogenic signals) have been reported. This would represent the first report of a signal produced by an organized geophysical Martian event; in this case a dust devil. As such, this has intrinsic interest, it provides data that complements theoretical work on the subject, and should be published. In addition, there is the potential that such measurements can inform studies of the meteorology of, and particle transport in, the Martian planetary boundary layer.

My primary concern is with the repeated use of the terms "sound" and "acoustic". While a microphone responds to pressure fluctuations, not all pressure fluctuations are acoustic. Acoustic fluctuations are propagating compression waves and are characterized by a dispersion condition relating the spatio-temporal wave-number and frequency. By contrast, the pressure signal recorded during the passage of a dust devil is a local, non-propagating signal related directly to the fluid motion of the vortex. While there could well be a propagating component that radiates away from the dust devil it is likely very small compared to the direct signal and, in any event, was not observed. The signals that are interpreted as from sand grain impacts might well be sound generated by the impacts nearby that then propagated to the microphone. I do not believe their interpretation of the spike at about 4.7 seconds seen in the spectrogram in Fig. 1. To me it looks like something, such as a pebble, could have smacked into the device generating a spike, essentially producing something close to the impulse response function of the microphone (including possible effects of the microphone mount, as suggested by the authors).

In part 7 of the methods section the authors state that wind noise is dominated by the intrinsic turbulence of the atmospheric flow. This depends on frequency and on the nature of the microphone housing and environment. An old paper by Raspert and Morgan is quoted. The authors should consult the more recent series of papers (from 2010 through 2020) by Raspert and coworkers. In particular, the influence of the sensor housing on the flow also produces turbulence. Regardless, the intrinsic turbulence levels, and the sensor induced turbulence levels do scale with mean wind speed in a way that can be calibrated to a particular location, sensor mounting, and frequency band. Thus the authors are correct that wind noise levels can be used to estimate wind speed.

My recommendation is that the authors revise the paper to indicate when they are measuring local pressure fluctuations as opposed to acoustic signals.

We are extremely pleased that the reviewers found our paper compelling, exciting and intrinsically interesting. Our detailed replies to each of the reviewers' comments are provided in blue below.

REVIEWER COMMENTS

Reviewer #1 (Remarks to the Author):

"The Sound of a Martian Dust Devil" by Murdoch et al., presents a novel investigation of a suite of measured characteristics of a dust-laden convective vortex impinging upon a craft on the surface of Mars, in this instance the Perseverance Rover. First-of-their-kind acoustic measurements of wind and sand grain impacts are described, resulting in derived vortex wind speed and grain impact quantification, which enhance interpretation of other more traditional near-surface atmosphere measurements and suggests that future measurements and analyses might offer the ability to quantify grain flux, which is an integral factor in establishing the martian dust cycle.

The manuscript presents a compelling description of and correlations between a number of environmental measurements diagnostic of the directly detected vortex characteristics (translation speed, vortex rotational wind speeds, atmospheric pressure deficit magnitude, gas temperature, atmosphere dust load) and derived vortex characteristics (diameter, height). Acoustic data provide novel sounds of a dust devil in the form of both lower frequency (10s of Hz) signals identified as diagnostic of wind signals, and higher frequency (few thousand Hz) measurements characteristic of grain impacts upon rover components.

It is exciting to see the capability of the suite of Perseverance instrumentation to provide such a thorough representation of conditions present with a martian dust devil, and very compelling to see the valuable investigative opportunities these data offer as presented in this manuscript.

I provide a few minor comments/thoughts about the manuscript's contents which the authors' might consider addressing to improve the experience for the reader.

1. The Perseverance RDS measurements are introduced (text line 85, 115, 151, ...) prior to the 'Radiative Dust Sensor' being described on text line 321; it would aid the reader less familiar with the MEDA instrument suite to introduce the Radiative Dust Sensor prior to or contemporaneously with the initial invocation of the RDS measurements

The definition of the RDS sensor and associated reference have now been moved earlier in the text to the first mention of RDS. Thanks.

2. In regard to the prominent acoustic signal occurring at the 4.6 second moment, indicated to be the sound signal of the impact of a large sand grain: is there any laboratory work available to estimate the size/mass of the grain required to produce a signal of such magnitude? With this impact occurring when the rover was within the 'calm' vortex interior (within the radius of maximum rotational winds), and the possible small magnitude wind speeds experienced there, is there some lower limit to gran mass than can be inferred (with appropriate assumptions such as the distance of the impact from the microphone)?

We would have very much liked to do exactly as the reviewer suggested and make a quantitative estimate of the distance to the impact and energy of the particle at impact. However, the lab work necessary to draw such conclusions does not currently exist, although we have mentioned it in the conclusions of the paper. We hope it will be part of future work.

3. Section 6 of the METHODS section possibly suggests that time-of-sol thermal inertia values are relevant in addition to time-of-sol dependent albedo. It might help the reader if the presentation of the 425 +/- 25 SI unit value of the thermal inertia was indicated to be representative of the thermal inertia during the entirety of the 26 sols Perseverance was at this location, if this is a correct statement

The reviewer has correctly understood: 425 +/- 25 SI unit value of the thermal inertia is representative of the thermal inertia during the entirety of the 26 sols Perseverance was at this location. The text in METHODS 6 has been modified to make this more obvious.

4. There is redundancy of the information presented in Figures 1d, 5b, and Supplementary Figure 1 (which seems to be referenced as "Fig. 1" on text line 293; Figure 5 provides a helpful visual comparison between the measured vortex pressure signature, the derived acoustic spectrum and other characteristics of the acoustic environment; however, Supplementary

Figure 1 does not add any new information compared to Figure 1 and the removal of Supplementary Figure 1 would not harm the manuscript's presentation. Additionally, Supplementary Figure 1 is not referenced within the main body of the manuscript nor in the METHODS Section. [And, Supplementary Figure 2 is not referenced within the manuscript main body or within the METHODS section.]

Supplementary Figure 1 has been removed. Supplementary Figure 2 (now Supplementary Figure 1) is now referenced in METHODS Section 2. There was an error in the reference previously, thank you to the reviewer for spotting this.

5. The graphed lines in the top four frames of Supplementary Figure 5 are very faint and could be difficult for the reader to see sufficiently

This top four frames of this figure (now Supplementary Figure 4) have been improved.

Reviewer #2 (Remarks to the Author):

This paper reports on the pressure signal recorded on the Mars Perseverance SuperCam microphone during the passage of a dust devil. Prior to this, only measurements of pressure fluctuations due to turbulence (wind noise) and signals associated with the Mars mission (anthropogenic signals) have been reported. This would represent the first report of a signal produced by an organized geophysical Martian event; in this case a dust devil. As such, this has intrinsic interest, it provides data that complements theoretical work on the subject, and should be published. In addition, there is the potential that such measurements can inform studies of the meteorology of, and particle transport in, the Martian planetary boundary layer.

My primary concern is with the repeated use of the terms "sound" and "acoustic". While a microphone responds to pressure fluctuations, not all pressure fluctuations are acoustic. Acoustic fluctuations are propagating compression waves and are characterized by a dispersion condition relating the spatio-temporal wave-number and frequency. By contrast, the pressure signal recorded during the passage of a dust devil is a local, non-propagating signal related directly to the fluid motion of the vortex. While there could well be a propagating component that radiates away from the dust devil it is likely very small compared to the direct signal and, in any event, was not observed.

We thank the reviewer for raising this point. The reviewer is entirely correct that the microphone does not measure the pressure 'dip' generated by the vortex itself, but rather the microphone signal that is caused by the vortex winds. We have stated clearly (Line 76) that the microphone "records air pressure fluctuations" and that "the microphone can be considered as a high frequency wind speed sensor capable of probing pressure variations associated with very short term wind speed fluctuations" (Line 80). See further details in response to the last question below.

The signals that are interpreted as from sand grain impacts might well be sound generated by the impacts nearby that then propagated to the microphone. I do not believe their interpretation of the spike at about 4.7 seconds seen in the spectrogram in Fig. 1. To me it looks like something, such as a pebble, could have smacked into the device generating a spike, essentially producing something close to the impulse response function of the microphone (including possible effects of the microphone mount, as suggested by the authors).

The presence of the acoustic echo (destructive interference) cannot be generated by mechanical transmission. This is due to the reflection of the acoustic wave on the base of the microphone mount and is seen in the acoustic signals generated by the SuperCam LIBS instrument. Therefore, we are confident that the large impact does indeed generate an acoustic signal on the microphone, rather than a mechanical transmission through the structure. Also, it would be very unexpected for the vortex to be capable of lifting pebbles; we don't expect anything larger than sand grains to be lifted by such events.

However, for the smaller impacts, we agree with the reviewer that without the necessary energy at the frequency range in which the echo is visible (~6 kHz), it is difficult to claim that the transmission is purely acoustic for the smaller impacts. Nevertheless, our estimates of grain number density based on peak detection in the microphone signal hold whether the microphone measured signal is of pure acoustic origin, and also in the case of mechanical transmission through the structure. We have reworded the text slightly to explain this uncertainty (see Lines 182 – 186). Thank you for raising this point and giving us the opportunity to improve our explanations.

In part 7 of the methods section the authors state that wind noise is dominated by the intrinsic turbulence of the atmospheric flow. This depends on frequency and on the nature of the microphone housing and environment. An old paper by Raspert and Morgan is quoted. The authors should consult the more recent series of papers (from 2010 through 2020) by Raspert and coworkers. In particular, the influence of the sensor housing on the flow also produces turbulence. Regardless, the intrinsic turbulence levels, and the sensor induced turbulence levels do scale with mean wind speed in a way that can be calibrated to

a particular location, sensor mounting, and frequency band. Thus the authors are correct that wind noise levels can be used to estimate wind speed.

As the reviewer points out, regardless of the mechanism for the wind noise generation we can use the observed sensitivity as an in-situ calibration. This is the basis for our analysis so we have updated the METHODS 7 section to reflect this and make it clearer to readers what we depend on. We have also updated the references and text to indicate the mechanisms for wind noise and how this relates to our calibration.

My recommendation is that the authors revise the paper to indicate when they are measuring local pressure fluctuations as opposed to acoustic signals.

Thank you for this good recommendation. We have tried to clarify the text where necessary.

REVIEWER COMMENTS

Reviewer #1 (Remarks to the Author):

The revised manuscript very adequately addresses my prior comments motivated by the original submission, and the well targeted review comments provided by another reviewer have also in my opinion been sufficiently addressed.

Reviewer #2 (Remarks to the Author):

The authors have misunderstood my comments and have therefore not responded to them.

1) The authors' statement "The reviewer is entirely correct that the microphone does not measure the pressure 'dip' generated by the vortex itself, but rather the microphone signal that is caused by the vortex winds" is false, and not at all what I meant. The microphone does measure the pressure dip generated by the vortex. I suppose they misinterpreted my statement that this pressure dip is directly tied to the fluid motion. Fluid acceleration and pressure gradients are by necessity coupled: the pressure gradient is the force that produces the acceleration. My point is that the microphone has measured a band passed version of that pressure dip. This is a near-field effect, not a propagating acoustic wave, ie. it is not sound. It would be extremely interesting to have seen an acoustic signal radiated from the vortex and detected at some distance from the vortex. No evidence of such an acoustic signal is being reported. This does not make the paper any less interesting, it just makes the statements about sound from the vortex false. It is a direct measurement of the pressure (some might call it quasi-static) associated with the passing vortex and not in any way related to the wind-noise generation, which seems to be the way my comment has been interpreted. Indeed, I would hope that the title of the paper would be changed to something like "Direct Detection of the Pressure Field Associated with a Martian Dust Devil".

2) My comments about the signals seen on the microphones from dust particles were also misunderstood, probably because I was not clear enough. Referring to Figure 1 from a recent paper (Maurice, Sylvestre, et al. "In situ recording of Mars soundscape." *Nature* 605.7911 (2022): 653-658) one sees that the microphone response has a sharp dip at about 6 kHz. This is reflected in the spectrogram shown in Figure 1 of the current paper. Regardless of the origin of this dip it is a feature of the microphone system frequency response. It is likely, as the authors claim, that this is a feature of the acoustic response, and not the mechanical response, however, my question is (and I admit

that it was poorly framed in my review) what caused that large broad-band signal at about 4 and 2/3 seconds? It's clearly quite different from the other grain "impacts".

REVIEWER COMMENTS

Authors' comments in blue.

Reviewer #1 (Remarks to the Author):

The revised manuscript very adequately addresses my prior comments motivated by the original submission, and the well targeted review comments provided by another reviewer have also in my opinion been sufficiently addressed.

Jim Murphy

We are glad Reviewer #1 found our revisions satisfactory.

Reviewer #2 (Remarks to the Author):

The authors have misunderstood my comments and have therefore not responded to them.

1) The authors' statement "The reviewer is entirely correct that the microphone does not measure the pressure 'dip' generated by the vortex itself, but rather the microphone signal that is caused by the vortex winds" is false, and not at all what I meant. The microphone does measure the pressure dip generated by the vortex. I suppose they misinterpreted my statement that this pressure dip is directly tied to the fluid motion. Fluid acceleration and pressure gradients are by necessity coupled: the pressure gradient is the force that produces the acceleration. My point is that the microphone has measured a band passed version of that pressure dip. This is a near-field effect, not a propagating acoustic wave, ie. it is not sound.

We agree with the reviewer that there has been misunderstanding about the way (and the words in our response) that properly describe the vortex sensing by the microphone. The formulation as a band-pass filtering effect is actually the right way to explain the characteristics of the time series recorded by the microphone. Nevertheless, we think it is important to distinguish the low frequency content, associated with the large scale of the vortex, from the high frequency content, even if all types of pressure fluctuations are equally linked to fluid acceleration from a mechanical point of view. The pressure drop at the spatial center of the vortex (that is what causes the circular motion) yields a quasi-Lorentzian dip in the pressure time series recorded at a given location. The characteristic duration of the latter scales with the ratio of distance (vortex diameter combined with miss distance) to the advection velocity. Differentiation of this pressure time series, or equivalently measurement by a high- or band-passed filtered pressure sensor such as those used to study infrasound, yields a bipolar pulse. Coauthors of this study first reported such signals (ref 31 – see below). An individual vortex passage shows as a single down-up-down ('heartbeat') signal a few tens of seconds long in these data (0.02 Hz to 10 Hz), with some higher-frequency noise superposed (presumably due to turbulent wind fluctuations, which form the core of the present paper).

Since the pressure dip (Fig 2b in the present paper) lasts only 10s, the pressure-derivative signal described by the reviewer would analogously be a single down-up-down infrasound (~ 0.1 Hz) signal. Our microphone has negligible sensitivity at such a frequency. This information is provided in the last sentence of the introduction: “Convective vortices also generate acoustic waves in the infrasound band (< 20 Hz), but these signals are outside the frequency range of the SuperCam Microphone”.

Our testing of the microphone in wind tunnels (ref 36) and experience with similar audio microphones in the field (see Supplemental figure 2) and intercalibration in-situ with wind sensors on the Mars rover (see Supplemental figure 7) confidently lead us to attribute the tens-to-hundreds of Hz audible signals to turbulent pressure fluctuations (see also ref 37).

Finally, our microphone records the “sound” of the vortex through the turbulent fluctuation caused by the interaction of the large scale phenomenon with the microphone neighbourhood. Thus, it seems to us comparable to the way in which a human ear reacts to strong winds (but not fully analogous given the role of the auditory canal), and this is the reason we refer to it here as a sound: what we can hear, but not necessarily/essentially a propagating wave.

It would be extremely interesting to have seen an acoustic signal radiated from the vortex and detected at some distance from the vortex. No evidence of such an acoustic signal is being reported.

Indeed, it has been an aspiration of ourselves and others for some time to detect radiated acoustic (audible or infrasound) signals from dust devils. One of us (RL) is party to a NASA-supported effort to detect and track dust devils this way. We have not reported that such radiated signals have been detected in this paper – we attribute the microphone signals to turbulent pressure fluctuations generated by the wind field associated with the dust devil interacting with the microphone directly, and/or with nearby rover structures.

This does not make the paper any less interesting, it just makes the statements about sound from the vortex false. It is a direct measurement of the pressure (some might call it quasi-static) associated with the passing vortex and not in any way related to the wind-noise

generation, which seems to be the way my comment has been interpreted. Indeed, I would hope that the title of the paper would be changed to something like "Direct Detection of the Pressure Field Associated with a Martian Dust Devil".

We assert that our microphone is not sensitive to such a low-frequency quasi-static signal. As described above, we therefore respectfully retain our original interpretation of the data.

2) My comments about the signals seen on the microphones from dust particles were also misunderstood, probably because I was not clear enough. Referring to Figure 1 from a recent paper (Maurice, Sylvestre, et al. "In situ recording of Mars soundscape." *Nature* 605.7911 (2022): 653-658) one sees that the microphone response has a sharp dip at about 6 kHz. This is reflected in the spectrogram shown in Figure 1 of the current paper. Regardless of the origin of this dip it is a feature of the microphone system frequency response.

It is likely, as the authors claim, that this is a feature of the acoustic response, and not the mechanical response, however, my question is (and I admit that it was poorly framed in my review) what caused that large broad-band signal at about 4 and 2/3 seconds? It's clearly quite different from the other grain "impacts".

The drop of the sensitivity around 6kHz results from the destructive interference at the position of the microphone capsule between a direct incident sound wave and the reflected one from the Supercam body a couple of cm behind the microphone tip. Thus, it is a feature of the instrument structure nearby, rather than the microphone transducer itself.

We agree that we did not give a definitive interpretation of the distinction of this event from the rest of the events, and we are planning laboratory studies to better understand it. The event (plotted in Fig 5d) is a loud and broadband event. The fact that the spectrum of this event has the 6 kHz notch leads us to conclude that the sound has been reflected from the front surface of the SuperCam Mast unit, in order for the interference to occur. Given the fact that this signal is unique (nothing similar has been seen in any other SuperCam microphone recordings), and the timing of this event (within the second burst of grain impacts), we suggest this signal is a particularly close impact, or the impact of a much larger particle.

The cloud of signals at ~3 kHz labeled 'grain impacts' in figure 1 are the familiar saltation 'hiss' very similar to those reported by Ellis et al. that are direct impacts on a microphone diaphragm. Their spectra do not show the 6kHz feature, thus we cannot determine if the signals are acoustic or mechanical in origin. However, given the large acoustic (and mechanical) attenuation we infer that they occur close to the microphone.

We have updated the section on 'Quantifying the vortex grain loading' taking into account the reviewer's pertinent remarks, for which we are grateful to the reviewer.

REVIEWERS' COMMENTS

Reviewer #2 (Remarks to the Author):

I think this paper should be published. It is acceptable to me as is.